# EXPLANATION USING SIMULATION

## ABSTRACT

In safety-critical domains, such as industrial systems, the lack of explainability in predictive 'black-box' machine learning models can hinder trust and adoption. Standard explainability techniques, while powerful, often require deep expertise in data analytics and machine learning and fail to align with the sequential, dynamic nature of data in these environments. In this paper, we propose a novel explainability framework that leverages reinforcement learning (RL) to support model predictions with visual explanations based on dynamical system simulation. By training RL agents to simulate events that require prediction, we use these agents' critics to make classifications. Next, we employ the actors of the RL agents to simulate the potential future trajectories underlying these classifications, providing visual explanations that are more intuitive and align with the expertise of industrial domain experts. We demonstrate the applicability of this method through a case study involving monitoring a small industrial system for cyberattacks, showing how our framework generates actionable predictions that are supported with visual explanations. This approach aims to bridge the gap between advanced machine learning models and their real-world deployment in safety-critical environments.

## 1 INTRODUCTION

With the increasing volume of data generated in industrial environments and the growing complexity of managing these systems, deep learning has become a promising tool to automate or support decision-making for industrial domain experts (Wang et al., 2022). A key function of deep learning in these settings is predictive analytics—making classifications to forecast future events, which can assist operators in planning for critical decisions. Despite the potential, the adoption of deep learning in industrial environments faces a major challenge: explainability. While deep learning models have demonstrated high reliability and accuracy, their 'black-box' nature has put into question their trustworthiness and limited their adoption in industries that must adhere to stringent regulatory standards (Ahmed et al., 2022). In safety-critical industries, transparency in decision-making is often mandated by regulations.

Several methods, notably model-agnostic techniques like Local Interpretable Model-agnostic Explanations (LIME) (Ribeiro et al., 2016) and SHapley Additive exPlanations (SHAP) (Lundberg, 2017), have been utilized to provide a degree of explainability for deep learning models. However, these techniques require a level of expertise in data analytics that most domain experts lack, making them difficult for industrial practitioners to interpret and apply effectively. Instead, many of these domain experts rely on time-series analysis, which involves observing data sequences over time to understand system behavior, predict future trends, and make informed decisions (Fatima & Rahimi, 2024). As the volume of data grows, this manual analysis becomes increasingly challenging, creating a greater need for deep learning systems that can support decision-making while remaining interpretable.

In this paper, we propose a novel approach that generates and explains deep learning-based classifications for industrial dynamic systems by providing time-series explanations that align with the workflows and practices of domain experts. Our approach leverages reinforcement learning (RL) agents to model the sequences of events that lead to key events that domain experts need to predict. Next, we demonstrate the use of the RL agents' state-action values or state values—generated by the agents' deep neural network-based critics—to make predictive classifications, and the learned action policies to simulate the sequence of events that lead up to those outcomes and underlie the classifi-

cation. This simulation offers an visually intuitive, dynamic explanation of how the model reaches its decisions that is more aligned with the skills and practice of industrial domain experts. Consequently, this method stands to enhance trust in artificial intelligence (AI), facilitating its adoption in industries where safety, reliability, and transparency are of paramount importance.

## 2 RELATED WORK

**Explainability**: Post-hoc explainable AI (XAI) methods, designed to explain the predictions of trained black-box models, can be broadly categorized into *model-specific* and *model-agnostic* approaches (Minh et al., 2022). Model-specific methods are tailored to a particular class of models, utilizing their internal structure to generate explanations. For instance, methods such as Layer-wise Relevance Propagation (LRP) (Montavon et al., 2019), DeepLIFT (Shrikumar et al., 2017), heatmaps (Payer et al., 2019), saliency maps (Adebayo et al., 2018), GradCAM (Selvaraju et al., 2020) backpropagate gradients from the output to the input layers in neural networks to visually emphasize regions in the input that have the greatest contribution to the model's decision. Model-agnostic methods, on the other hand, do not rely on the internals of a specific model. Instead, they aim to provide explanations based purely on input-output behavior. SHAP (Lundberg, 2017) is one such method, which explains the contribution of each input feature to the prediction using game-theoretic principles. LIME (Ribeiro et al., 2016) approximates the black-box model locally using a simpler, interpretable model and uses the coefficients of this simpler model to explain the relative importance of each feature to the prediction. More recently, textual justification methods have emerged as an explainability approach (Shi et al., 2018; Sabol et al., 2020; Musto et al., 2021; Aminimehr et al., 2024; Hartmann et al., 2022). These methods generate natural language explanations of model predictions, aiming to make the reasoning accessible to general users. Musto et al. (2021) showed that text justification can generate user-friendly explanations that improve transparency and enhance user trust and satisfaction in the context of recommendation systems. A notable trend is that textual justification methods have been primarily proposed for applications such as recommendation systems, medical diagnosis, and stock prediction, highlighting the effectiveness of this explanation approach for domain experts in specialized fields. The motivation for our work aligns with that of textual justification, but offers a different medium of explanation tailored to domain experts in industrial domains: time-series plots.

**Explainable RL**: While sharing the same tools, the field of explainable RL (Wells & Bednarz, 2021; Heuillet et al., 2021) typically focuses on interpreting the reasoning, actions, and decision-making processes of RL agents after they have learned a task. In contrast, our approach does not aim to explain the internal decision-making of the agents themselves. Instead, we leverage the agents' learned behaviors to explain how specific predictions could unfold in the system. The goal is not to understand the agent's reasoning but to use its actions to simulate potential future events, thereby providing a clearer explanation of the predicted outcomes.

**RL in Industry:** RL research is pervasive across industrial sectors, including power systems (Zhang et al., 2019), autonomous driving (Aradi, 2020), smart cities (Ullah et al., 2020), and manufacturing (Wang et al., 2021).

## 3 CONCEPT

RL agents learn to make decisions by interacting with a dynamical system environment, aiming to maximize cumulative rewards (or minimize cumulative penalties). In an actor-critic RL architecture, the agent consists of two key components: the actor and the critic. The critic estimates a state value: the expected return (cumulative future rewards) from a given state $S$ under a policy $\pi$, expressed as

$$V^\pi(S) = \mathbb{E}\left[\sum_{i=0}^{\infty} \gamma^i R_{t+i} \mid S_t = S\right], \tag{1}$$

or, alternatively, a state-action value (or Q-value): the expected return starting from state $S$, taking action $A$, and then following policy $\pi$, expressed as

$$Q^\pi(S, A) = \mathbb{E}\left[\sum_{i=0}^{\infty} \gamma^i R_{t+i} \mid S_t = S, A_t = A\right]. \tag{2}$$

$\gamma \in [0, 1)$ is a discount factor determining the importance of future rewards.

Through training, the critic learns to map the data, collected as observations, of the environment to a numerical value that predicts the event the RL agent is trained to achieve. For instance, if an RL agent is trained to inject malicious disturbances or manipulate machinery in an industrial setting, the critic learns to map system data to a state value that predicts when these manipulations could lead to equipment damage or system failure. Alternatively, in an autonomous driving simulation, where an RL agents affect the external vehicle's motion, the critic learns to anticipate risky driving situations based on the car's external environment data.

In our research, we extend the critic's state value to generate both numerical and categorical values that provide predictive classifications based on system data.

The actor, on the other hand, dictates the actions taken by the agent. Since the actor learns to simulate the sequence of actions that lead to the predicted event, we leverage the actor to provide explainability. Specifically, the actor is used to simulate the sequence of events leading to a classification, providing supporting visual explanation.

Figure 1 illustrates our conceptual framework. We train multiple RL agents, each designed to cause a different event that requires prediction. Each agent's critic processes observations from the production system and feeds its state-action (or state) value to a function that maps it to a numerical score representing the event's impact. These mappings may include the probability of the event, which in negative impact cases, translates to a measure of risk—commonly defined as the product of impact and likelihood. The outcome with the highest impact (or risk) determines the classification. When prompted for an explanation, the actor corresponding to the highest impact generates the action sequence that leads to the predicted outcome. Next, the system's simulated behavior under the actor's actions is visualized as a time-series plot, allowing the user to understand the system's future trajectory underlying the classification.

To demonstrate the advantages of our approach, particularly in safety-critical systems, we consider the scenario of a small, localized electric microgrid targeted by cyberattackers. In this scenario, a predictive analytics system must monitor the microgrid's data and output a classification that reflects the potential impact of cyberattacks manipulating a generator's load-frequency control. Maintaining a stable power frequency—at 60 Hz in North America—is critical for the proper functioning of the grid. Deviations cause power flicker (experienced as lights rapidly dimming and brightening), which can damage sensitive equipment by forcing it to operate outside of its designed-frequency range. Power system equipment is expensive, and significant frequency anomalies are typically detected by protection devices, which isolate the equipment to prevent their damage. Consequently, by strategically compromising control, cyberattackers could potentially isolate generators and cause power blackouts. The dangers posed by such cyberattacks have been the topic of numerous research studies, surveyed in (Mohan et al., 2020).

We choose this scenario for several important reasons. First, power systems are increasingly the target of sophisticated cyberattacks, with substantial evidence pointing to nation-state actors involved in cyber espionage (Hjortdal, 2011) and attacks on electric grids, as demonstrated in the attacks on Ukraine's grid in 2015 (Case, 2016) and 2016. Power systems are also massive in scale in complexity, and stand to significantly benefit from incorporating deep learning-based methods for monitoring and risk assessment. Additionally, power systems domain experts cannot be expected to have extensive data analytics expertise. Second, the lack of data on cyberattacks makes it challenging to predict threats from historical data. RL becomes a viable tool here, as it can generate simulated data to anticipate the impact of unknown cyberattacks and help detect them when they occur. Finally, cyberattacks often involve subtle and strategic manipulations of the system. The critic in RL can help uncover these complex failure modes, making it an effective approach for predicting and explaining potential risks in critical infrastructure systems.

## 4 METHOD

Considering a dynamical system expressed as:

$$\dot{\boldsymbol{x}} = g(\boldsymbol{x}, \boldsymbol{u}, \boldsymbol{A}) \tag{3}$$

$$\boldsymbol{S} = h(\boldsymbol{x}, \boldsymbol{u}, \boldsymbol{A}) \tag{4}$$

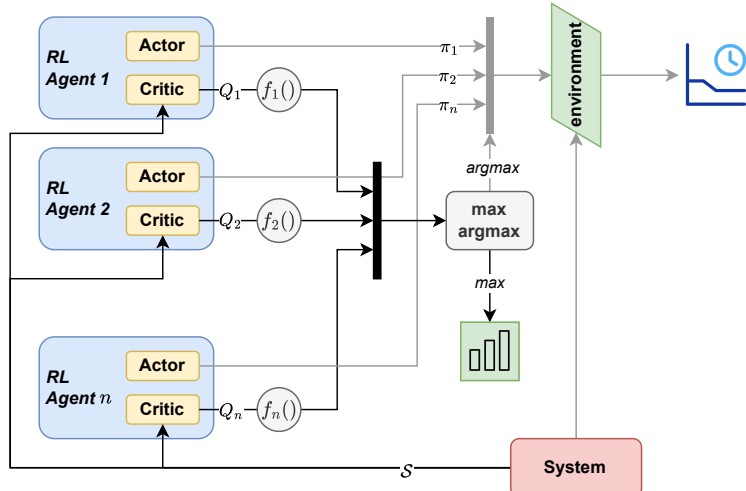

Figure 1: Concept illustrating $n$ critics providing classifications and $n$ actors supporting with explanations.

where $\boldsymbol{A} = \{A_1 \cup A_2 \cup \cdots \cup A_n\}$ represents the set of actions that $n$ RL agents can input into the system, and $\boldsymbol{S} = \{S_1 \cup S_2 \cup \cdots \cup S_n\}$ denotes their respective observations. $\boldsymbol{u}$ denotes inputs not generated by the RL agents, and $\boldsymbol{x}$ represents the state of the dynamical system. Each RL agent is trained to achieve specific outcomes that domain experts need to predict. Depending on the use case, training can involve different paradigms, including single-agent or multi-agent reinforcement learning. This formulation allows for both single-agent and multi-agent RL setups.

Once each RL agents converges on an action policy, each agent $i$ will have trained deep neural networks for both its actor and critic. The actor network $\pi_i$ maps the agent's observations to actions:

$$\pi_i : S_i \rightarrow A_i \tag{5}$$

which selects actions that, when applied greedily, aim to maximize the agent's expected cumulative reward. The critic network estimates the value of the agent's state, either as a state-value function:

$$\mathcal{C}_i : S_i \rightarrow V_i \tag{6}$$

or as a state-action value function (Q-value):

$$\mathcal{C}_i : (S_i, A_i) \rightarrow Q_i \tag{7}$$

The critic's values are dependent on the reward function defined for each RL agent. These values are then mapped to a measure of predicted impact or risk using the following function:

$$f_i(V_i(S_i)) = P_{\text{likelihood},i} \cdot f_{\text{impact},i}(V_i(S_i)) \tag{8}$$

where $f_{\text{impact},i} \geq 0$ is a non-negative function that increases with the predicted impact of the outcome, and $P_{\text{likelihood},i}$ (optional) represents the probability of the outcome, which allows risk estimation. Alternatively, when using the Q-value, the impact or risk is computed as:

$$f_i(Q_i(S_i, A_i)) = P_{\text{likelihood},i} \cdot f_{\text{impact},i}(Q_i(S_i, A_i)) \tag{9}$$

At runtime, each agent's critic receives real-time observations from the production system, and the agent with the highest predicted impact or risk score is selected:

$$\{j, f_j(S_j)\} = \arg\max_i \{f_i(S_i)\}^{i \in \{1, \cdots, n\}} \tag{10}$$

For better comprehension, the score can also be categorized based on predefined thresholds or classes. When an explanation is required for a classification, the policy $\pi_j$ of the agent with the

highest impact or risk score is used to simulate the sequence of actions from time $t$ up to a pre-defined horizon $t + T$ or until the outcome has been realized in the simulation, i.e.,

$$A_j = \pi_j(S_j) \tag{11}$$

$$A_i = \mathbf{0} \text{ for } i \in \{1, \cdots, n\} \backslash j \tag{12}$$

The system state is then presented as a time-series plot, providing the domain expert with a visual explanation of the classification or predicted impact or risk score.

### 4.1 EXPERIMENT

We present a case study of a small microgrid, adapted from Mohamed & Kundur (2024). The microgrid is equipped with a security system that monitors the frequency and its rate-of-change (derivative), providing a classification that represents the vulnerability of the system to potential cyberattacks. When a high-urgency classification is detected, it signals the likelihood of an ongoing cyberattack that may be compromising the system. In such a scenario, a critical decision—such as isolating compromised communication channels—must be made to mitigate the attack. However, communication between system components is vital for the safety and stability of the microgrid, so this decision must be based on a highly trustworthy classification, triggered only under real threat to prevent further system damage.

In this experiment, we train RL agents to simulate cyberattacks aimed at compromising the system and forcing it into blackout conditions. The RL agents are tasked with learning strategies that satisfy several key objectives:

1. The cyberattacks must introduce disturbances to the generator's control system, manipulating its power output. An increase in power generation will lead to a rise in the microgrid's frequency, while a reduction in power generation will decrease the frequency.

2. Both the frequency and its rate-of-change are measured in per-unit (pu), with frequency normalized by the nominal 60 Hz. Similarly, the rate-of-change is divided by 60 Hz for a consistent measurement unit. Protective devices within the microgrid are programmed to activate when the frequency or its rate-of-change deviates from safe thresholds, typically 3% pu for frequency deviation and 5% pu for the rate-of-change.

3. It is desirable for the attack to keep the frequency deviations minimal to avoid triggering countermeasures. The more subtle the deviations, the harder it is to detect the attack. Therefore, the RL agents should aim to manipulate the rate-of-change of the frequency rather than the absolute frequency itself to ensure stealthier attacks.

To meet these objectives, we design a reward function that guides the RL agent to force large deviation in the rate-of-change of frequency while keeping the frequency deviation minimal. The reward function is defined as follows:

$$R_t = \left(\frac{s_2}{5\%}\right)^2 \cdot \max\left(0, 1 - \left(\frac{s_1}{3\%}\right)^2\right) + 30\{|s_2| > 5\%\} - 5\{|s_1| > 3\%\} \tag{13}$$

In this reward formulation, $s_1$ denotes the frequency deviation from nominal, and $s_2$ is the rate-of-change of the frequency. The agents observations are $S = (s_1, s_2)$. The first term in equation 13 encourages the agent to increase the rate-of-change of frequency while minimizing the frequency deviation. The second and third terms provide additional incentive: a large positive reward for pushing the rate-of-change outside the safe operating range of 5%, and a large penalty for large frequency deviations. Episodes terminate when the conditions in the second and third terms are satisfied.

Appendices A and B provide further details on the state-space representation of the dynamical system and the architecture and hyperparameters of the Deep Deterministic Policy Gradient (DDPG) agent used in the experiments.

## 5 RESULTS

Figure 2a illustrates the action sequence injected into the system by a RL agent. The effect is shown in Figure 2b, where the agent induces a resonance in the system, causing the rate-of-change of

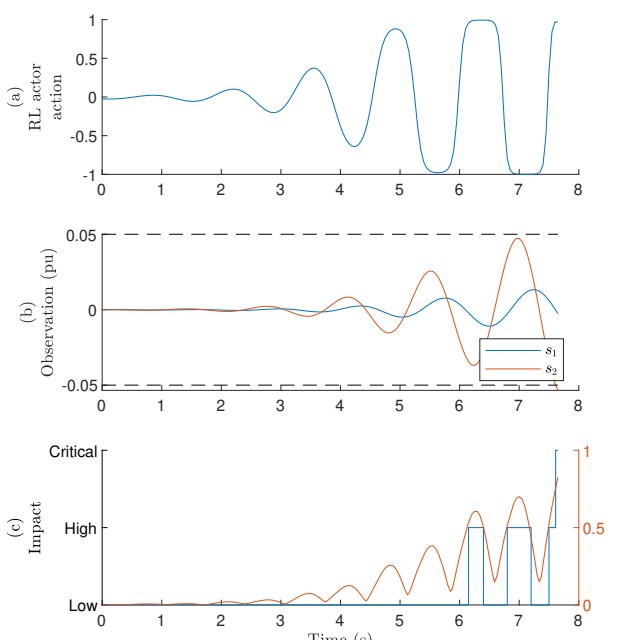

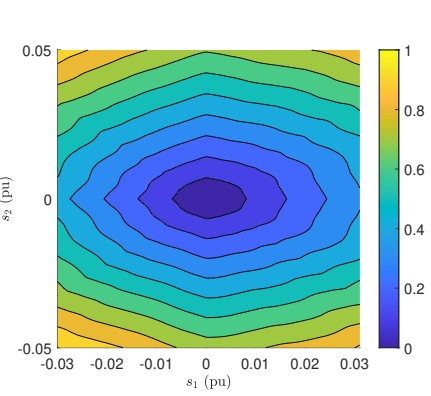

Figure 3: Impact score visualized as a function of the observations.

Figure 2: (a) action sequence generated by RL agent; (b) effect on system observations; (c) impact score and category.

frequency (depicted in red) to progressively increase until it breaches the lower threshold, marked by the lower dashed horizontal line. During training, this RL agent consistently favored driving the rate-of-change below the lower safety threshold. To include contrasting behavior, we trained a second RL agent to manipulate the system such that the rate-of-change of frequency exceeds the upper safety threshold.

After training, we observed that the Q-values of the agents decrease (i.e., become more negative) as the system approaches failure. This behavior occurs because, near failure, the agent has fewer remaining timesteps to collect potential rewards, whereas, further from failure, the agent can accumulate rewards over a larger time window. Consequently, the Q-value is larger when the system is far from failure. Based on this, we defined the function mapping the Q-value to the impact score as follows:

$$f_1(Q(S, \pi(S))) = f_2(Q(S, \pi(S))) = \frac{e^{-Q(S,\pi(S))} - 1}{M} \tag{14}$$

where $M$ normalizes the impact corresponding to the Q-value near system failure. Specifically, $M$ is determined as:

$$M = e^{-\min\{Q(S,\pi(S))\}} - 1 \tag{15}$$

The forms of $f_1(\cdot)$ and $f_2(\cdot)$ ensure that the impact score is non-negative and increases as the system approaches failure. For simplicity, we do not weigh $f_1(\cdot)$ or $f_2(\cdot)$ over one another, nor do we consider likelihoods of events, i.e., $P_{\text{likelihood},i} = 1, i \in \{1, 2\}$.

Figure 3 visualizes the impact as a function of the system's observations. The impact $\mathcal{I}$ is calculated as:

$$\mathcal{I} = \max\{f_1(Q_1(S, \pi(S))), f_2(Q_2(S, \pi(S)))\} \tag{16}$$

Given the low-dimensional observation space, this visualization effectively captures the relationship between the system's state and the risk of cyberattacks. Intuitively, the plot shows that the impact

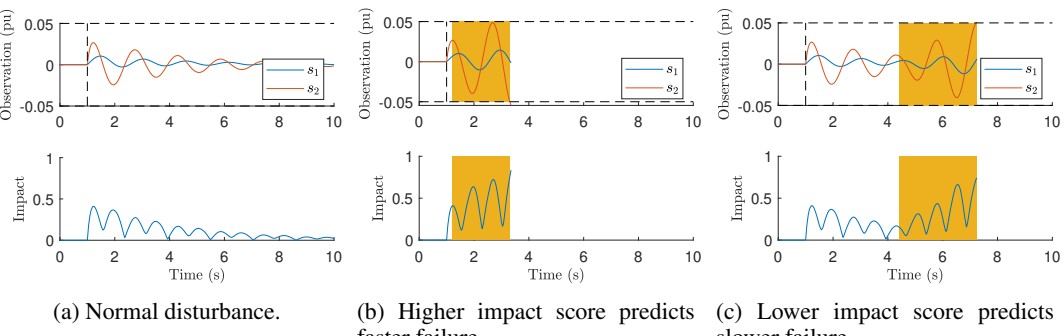

(a) Normal disturbance.

(b) Higher impact score predicts faster failure.

(c) Lower impact score predicts slower failure.

Figure 4: Explaining real-time classifications made by the RL agents' critics via simulations. The yellow zones represent the simulated portions.

increases as the rate-of-change of frequency approaches the 5% pu thresholds. We can further categorize the impact into three levels, which can be used for system alerts or state classification:

$$I \in [0, 0.5) : \text{Low; system is safe, attack unlikely}$$

$$I \in [0.5, 0.7) : \text{High; potential risk to the system, possible attack}$$

$$I \in [0.7, 1] : \text{Critical; imminent system failure, active attack}$$

Notably, the 'Critical' category alert to $s_2$—the rate-of-change of frequency—nearing the critical 5% pu threshold, signaling a high likelihood of system failure due to a cyberattack. Hence, action must be taken to mitigate attacks when the system is in 'High' category.

To illustrate, Figure 2c shows the evolution of the numerical impact value (in red) during the cyberattack, alongside the corresponding categorical impact levels (in blue). As the attack progresses, the impact value continues to rise. The attack starts at 0 seconds. Approximately 6 seconds after, the classification spikes to level 2, indicating a high likelihood of an ongoing cyberattack. Just before the 8-second mark, the classification briefly spikes to level 3, signifying imminent system failure right before the attack succeeds in failing the system.

To further illustrate the use of the classification system and its explainability during normal operation, Figure 4a shows the effect of a normal system disturbance on the observations $s_1$ and $s_2$. During this disturbance, the impact increases but remains within level 1, indicating that the system is safe. The impact value reaches its peak at 1.2 seconds, signaling a moment of relatively heightened vulnerability. The impact value suggests that a cyberattack initiated precisely at this point in time would pose a greater risk compared to one initiated later. We use our proposed method to verify and explain this. Figure 4b shows the actor's explanation of an attack that begins exactly at the moment when the impact value peaks, and continues through the period highlighted by the yellow zone. For comparison, Figure 4c illustrates the effects of an attack that starts later. By examining these figures side by side, it becomes evident that the system fails more quickly when the attack is launched at the instance of peak impact in Figure 4a. This demonstrates the system's heightened susceptibility to failure at that specific point in time, validating the predictive and interpretable utility of the proposed method.

## 6 DISCUSSION

**Accessibility of explanation through simulation:** One of the key strengths of this approach is its ability to offer explanations for classifications and predictions in an intuitive and accessible format, particularly for industrial domain experts. By leveraging simulations, this method aligns well with the existing expertise of these professionals, who are accustomed to monitoring time-series plots, events, and sequences. The simulations create a visual narrative that makes understanding the underlying dynamics more straightforward, thereby facilitating the integration of machine learning insights into industrial systems.

**Data generation in the absence of historical data:** Another advantage of this method is its capability to generate synthetic data for events that lack historical precedent. By employing multiple RL

agents, the system explores and predicts a diverse range of events, including those that are subtle or previously unknown. Moreover, this approach enables domain experts to iteratively interact with the RL environment, allowing them to introduce new test cases or expand the number of events requiring prediction. This adaptability is key to refining the predictive model over time, as experts can augment RL agents and environments with additional details as necessary.

## 6.1 CHALLENGES

**Expertise required for RL design:** Although our approach alleviates the need for deep technical expertise in explainability methods, we acknowledge the complexity involved in designing and training RL agents. This presents a potential barrier, as creating effective RL models demands specialized knowledge. However, the trade-off is that this complexity is primarily concentrated in the RL development phase, which occurs infrequently. In contrast, the resulting system, once in place, simplifies interpretation and understanding, a task that will be required far more often by a broader audience of users.

**The importance of historical data:** While our focus has been on the use of RL in generating predictions through simulation, we recognize the value of historical data in classification algorithms. Historical data provides critical insights into patterns that can inform decision-making. Although our method primarily emphasizes online RL learning in simulated environments, there is potential to use offline RL (Levine et al., 2020), which leverages previously collected datasets. A hybrid approach, combining both historical data and simulated environments, could further enhance the predictive models.

**Simulation environments:** A key limitation of relying on online RL is that it necessitates the availability of accurate dynamical models that can be incorporated into the RL environment. For many industrial systems, models form the foundation of system design and are readily available. Furthermore, with the rapid growth of digital twins and simulation software, domain experts will increasingly have access to models that closely mimic real-world environments. This progression will help bridge the gap between virtual simulations and real-world applications, making the adoption of our method more feasible.

## 7 CONCLUSION

In this paper, we proposed a novel direction for explaining deep learning model classifications through time-series simulations. We developed a real-time predictive system for dynamical environments, where multiple reinforcement learning (RL) agents are employed to make predictions and support these predictions with simulations that visualize the forecasted system behavior underlying the classification. The motivation behind this approach is to offer an alternative method for explainability in contexts where domain experts are more accustomed to analyzing time-series plots to understand system behavior and anticipate future outcomes.

This work paves the way for further exploration of RL-based explainability, particularly in settings where traditional post-hoc XAI methods fall short. We encourage future research to apply this framework to a range of industrial case studies, explore hybrid RL approaches that combine offline and online learning, and investigate the method in diverse RL neural network architectures to further enhance the potential of this method.

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

## A MICROGRID MODEL

The microgrid model is as follows:

$$
\dot{\boldsymbol{x}} = \begin{bmatrix}
0 & 0 & 0 & -(kB) & 0 & 0 \\
1/\tau_G & -1/\tau_G & 0 & -d/(\tau_G) & 0 & 0 \\
0 & 1/\tau_T & -1/\tau_T & 0 & 0 & 0 \\
0 & 0 & 1/M & -D/M & 0 & 0 \\
0 & 0 & 0 & 1/\tau_\omega & -1/\tau_\omega & 0 \\
0 & 0 & 1/(M\tau_\nu) & -D/(M\tau_\nu) & 0 & -1/\tau_\nu
\end{bmatrix} \boldsymbol{x}
$$

$$
+ \begin{bmatrix}
0 & 0 \\
0 & -k \\
0 & 0 \\
-1/M & 0 \\
0 & 0 \\
-1/(M\tau_\nu) & 0
\end{bmatrix} \boldsymbol{u} + \begin{bmatrix}
k \\
0 \\
0 \\
0 \\
0 \\
0
\end{bmatrix} \boldsymbol{A}
$$

Table 1: Model Data

| Parameter | Symbol | Value |
|---|---|---|
| AGC gain | $k$ | 3 |
| Droop gain | $d$ | 40 |
| Governor time-constant | $\tau_G$ | 0.08 |
| Turbine time-constant | $\tau_T$ | 0.45 |
| Generator inertia | $M$ | 6 |
| Damping constant | $D$ | 0.03 |
| Frequency sensors time-constants | | $\tau_\omega = \tau_\nu = 0.1$ |
| Control center frequency measurement gain | | $B = 1$ |

# B  RL Agent Architecture

Table 2: DDPG neural network architectures and hyperparameters

| Actor network | | |
| --- | --- | --- |
| **Layer** | **# of units** | **Hyperparameters** |
| Input | $2\ (s_1, s_2)$ | $M = 128$ |
| Normalization | 2 | $\alpha_\theta = 10^{-4}, \alpha_\phi = 10^{-3}$ |
| Fully-connected | 100 | $\gamma = 0.99$ |
| ReLU | | $\tau = 10^{-3}$ |
| Fully-connected | 50 | $N \sim \mathcal{N}(0, 0.3)$ |
| ReLU | | |
| Tanh (or Sigmoid) | | |
| Scaling | 1 | |
| Output | $1\ (A)$ | |

| Critic network | | | |
| --- | --- | --- | --- |
| **Layer** | **# of units** | **Layer** | **# of units** |
| Input | $2\ (s_1, s_2)$ | Input | $1\ (A)$ |
| Normalization | 2 | Normalization | 1 |
| Fully-connected | 100 | Fully-connected | 50 |
| ReLU | | | |
| Fully-connected | 50 | | |
| Addition | 50 | $\swarrow$ | |
| ReLU | | | |
| Fully-connected | 1 | | |
| Output | $1\ Q(s_1, s_2, A)$ | | |