# OpenReview forum: "Explanation using Simulation"
_ICLR.cc/2025/Conference — ICLR 2025 Conference Withdrawn Submission_

### Official Review · Reviewer_kGMC · 2024-10-24

**Soundness:** 2
**Presentation:** 2
**Contribution:** 2
**Rating:** 3
**Confidence:** 3

**Summary:**

The paper introduces a new explanation method for industrial systems using supervised learning by leveraging reinforcement learning. The method does not explain reinforcement learning agents but uses them to explain an industrial system via simulation and observing the outcome. The method is tested on a microgrid system with simulated cyberattacks on the system.

**Strengths:**

- The paper is relatively straightforward to read and it flows.
- The method seems novel to me. That said, I do not work on explaining time series.

**Weaknesses:**

I have divided the comments into general and more specific local once.

### General Comments
- Although I said it was straightforward to read it, I have a hard time understanding the correctness of the method. The paper has more than two pages left, which I believe the authors should use to flesh out the details.
- There is only one, very specific experiment. I have a hard time understanding how applicable this method is in general. I recommend running more experiments.
- I believe the paper should tell readers that the measure of risk that is talked about in Section 3 and 4 is actually manually designed much earlier. We only get to know that in Section 5.

### Specific Comments
- Section 2, Related Work: The related work mentions a lot of general explainability methods. I believe a stronger paper would highlight works that are closer to their own. Currently, the mentioned methods that are quite general, for example LIME, SHAP, Grad-CAM and so on.
- Page 2, Line 067: Not all the methods mentioned use gradient to create saliency maps. Please correct this factual error.
- Page 2, Line 091: The related work on industrial application is quite thin, even though this paper is motivated for industrial use.
- Page 2, Equation 1 and 2: Please indicate what we are sampling (trajectories), since it is an expectation. Also, R is never formally defined in the text.
- Figure 1: Please add a better caption to make it more descriptive. Ideally, I want figures to be self-contained.
- Page 3, Line 126: I am not sure what this is supposed to tell the reader. Please clarify.
- Page 3, Equation 3 and 4: Please describe what g and h are.
- Page 4: Equation 6 and 7: In Equation 1 and 2, V and Q are used as functions, but here they are used as codomains.
- Page 4: Equation 10: The notation used seems to be wrong.
- Section 3, Concept: This title is not very descriptive and the section is confusing since it does not contain one coherent theme. The first half seems to introduce background knowledge while the latter introduces experimental scenario? I would like to see a better organization of the paper.
- Section 4.1: I would prefer the experiments to be organized with the results and not in the method section.

**Questions:**

- Page 1, Line 011: What are "standard" explainability techniques?
- Page 1, Line 029: What are these "industrial environments"?
- Page 1, Line 041: What backs the claim that Lime and SHAP are too difficult to use for domain experts? Do domain experts really need a lot of training to use these methods?
- Page 3, Line 110: This definition of state value sounds wrong to me. Is this really what the value function tells us?
- Page 4, Line 186: Are there any features generated by the RL agent? Since we have some not generated by the RL agent. And how do RL agents generate features? Or is this referring to the next state because the RL agents are interacting with the environment, and the environment returns the next state.
- Page 4, Line 188: How does this statement "the formulation allows for both single-agent and multi-agent RL setups" give more insight into the method? Could the authors add more detail? And how can we verify this is true?
- Page 4, Equation 8 and 9: How does this go from state and state-action values to risk?
- I am doubtful about the usefulness of these explanations. I know that state values themselves are not interpretable, which is why there is a separate research field studying to understand RL agents. How is a transformation of these state values going to help understand these industrial systems?

---

### Official Review · Reviewer_RviZ · 2024-10-27

**Soundness:** 3
**Presentation:** 2
**Contribution:** 2
**Rating:** 3
**Confidence:** 4

**Summary:**

The paper proposes a method to explain dynamical system classifications using reinforcement learning. The method works by training RL agents to simulate future trajectories and provide visual explanations. A case study is provided in the cyberattacking domain.

**Strengths:**

I think overall the problem domain and application are extremely interesting, and the usage of RL to simulate future events and utilize that for classification makes a lot of sense. In general, I also found the writing and presentation to be carefully thought out and mostly friendly to readers.

**Weaknesses:**

For me, there are two main drawbacks of the paper:

1. The evaluation is not particularly convincing, as no clear application is shown of the method, it is a short case study showing some examples, but it is not obvious to me after reading the paper how this would benefit real users of the system, or engineers trying to e.g. debug it.
2. Again related to the evaluation, there is no baselines compared to, it's not even obvious that this would be better than random chance.

Typo on line 114 agents -> agent

RL in Industry paragraph needs some text explaining why you are citing all that work, and how it relates to yours.

**Questions:**

Please respond to my two critiques above.

---

### Official Review · Reviewer_dEVz · 2024-11-01

**Soundness:** 1
**Presentation:** 1
**Contribution:** 1
**Rating:** 3
**Confidence:** 4

**Summary:**

The paper proposes to use reinforcement leanrning for implementing simuations that help explaining model predictions.

**Strengths:**

NA

**Weaknesses:**

I found the paper a bit hard to read -- in particular understanding its novelty and contribution.
If I understood it correctly, RL is used to simulate different events in a system (e.g. power grid). However, I do not really understand how this helps to explain the predictions of a given model -- are you aiming for some kind of "What-If" explanation?

I think the paper would benefit from more structure. For instance, clearly state/define the problem and the proposed solution. If a framework is proposed, each step/part of it should be described in detail. Right now, too many things are mixed in the paper.


Minor:
Line 037: I think the EU AI act poses some requirements on transparency if the AI is applied in a safety-critical environment
Line 091: I think this section on "RL in Industry" can be dropped because it does not contain any relevant information.

**Questions:**

Experiments:
What exactly is the explanation?

---

### Official Review · Reviewer_R5sr · 2024-11-07

**Soundness:** 1
**Presentation:** 1
**Contribution:** 1
**Rating:** 3
**Confidence:** 2

**Summary:**

In this paper, the authors propose a method where they adopt actor-critic RL methods to model industry dynamical systems (e.g., risks posed by cyberattacks to a microgrid). Specifically, the authors walk through a case study where they define an RL environment, reward function, and "risk score" (a function of the reward) of an adverse event. They then define a simple classification model that thresholds this risk score to determine cybersecurity risk. The authors argue that domain experts can use visualizations of RL agents' state and "risk score" over time, to interpret the learned models.

**Strengths:**

* The authors' motivating use case – a scenario where one wishes to monitor the power frequency of an electric microgrid by classifying the risk it is under a cyberattack – is a well-motivated, real-world scenario.
* The authors' approach of modeling cybersecurity risk using a learned RL "critic" model is innovative, and it is clear to me that the simple thresholding classifier that they created is easy to visualize, and thus may be more "interpretable" to a decision-maker.
* In their Appendices, the authors provide detailed information about their microgrid model and RL agent architecture to promote reproducibility.

**Weaknesses:**

My primary critique of the paper is that in the way it is currently written, I struggled to understand the authors' methodological contribution as described in Sections 1, 3, and 4. I found both the language the authors used to describe their contribution and the notation used in their equations to be confusing. Figure 1 also could benefit from a more descriptive caption. Below, I pull out several of my key points of confusion, and when applicable, point to specific passages that contributed to my confusion.

* **Classification**: As early as possible in the draft, you should clarify how exactly you intend "use [RL agent's] critics to make classifications" – as hinted at in your abstract.
  * If my understanding is correct, the final "classification" model simply thresholds the "impact" score, which is defined as a function of the learned Q-value (reward function) of the critic. So at any given point in time, if the impact score (reward) is in the "high risk" range, it will be flagged.
The reward function is defined by the research team, to correspond to real-life "events" that you're trying to monitor – for example, a cyberattack by a bad actor.

When I was first reading the draft, I was very confused by the use of the terms "classification", "impact", and "event". For example, consider Section 3, where you first use these words when describing your approach:

> "Through training, the critic learns to map the data of the environment to a numerical value that predicts the event the RL agent is trained to achieve."

This sentence is confusing. What do you mean by "event" here? The "numerical value" = the reward. How are rewards tied to "event"s?

> "For instance, if an RL agent is trained to inject malicious disturbances or manipulate machinery, the critic learns to map system data to a state value that predicts when these manipulations could lead to equipment damage or system failure."

How can a state value, which is the expected return (a number), "predict when" an event will occur? Numbers can't make predictions.

* **Why multiple agents?**: I don't understand what the benefit is in having multiple distinct agents, as illustrated in Figure 1.
  * In Section 4, Equations 5-10, you use the index $i$ to imply that each agent will have its *own* reward function, "impact score", and thus its own optimal policy and Q-value. Can the authors add some text in this section intuitively explaining why this is? Is the idea that each agent can represent an explicit/unique set of "risks"?
  * But then, in Section 4.1 and Equation 13, it appears that the authors use the *same* reward function for both agents. When justifying why we need two of them, the authors state that they "trained a second RL agent [to include contrasting behavior]". I don't get it - what exactly do you mean by "contrasting behavior"?

I have several other critiques that are secondary to the above.
* **Weakness 2: Generalizability of method (simulation) to other contexts**.
  * To be able to use RL to make classifications, the authors had to construct an environment, reward functions (for multiple agents), and "impact functions" that corresponded to adverse events. The authors tell us what these things are for their context, but do not include much reflection in the text on _how_ exactly they made these decisions. What's unclear to me is whether it is feasible, easy, or even desirable to go through this process in another new context. The authors touch on this point a bit in the final "Challenges" section.
  * I think this fundamental methodological limitation – that the authors' proposed method requires that the modeler can formulate their task by defining an RL environment with a known reward function – should be surfaced as early as possible. The way the paper is written implies that the authors' methodology is applicable _in all settings_ where one may wish to do classification, when my gut sense is that this assumption will be restrictive in many such settings.

* **Weakness 3: Comparing to the wrong XAI baselines**.
  * The interesting part of the authors' work was that they were able to formulate their time-series anomaly detection task by constructing an "impact" function, using a critic model. Once you're able to measure impact/risk, the authors' proposed classifier – simple thresholding of the risk score – is naive. Of course a simple thresholding model is interpretable, given that you accept the premise that the "risk score" is interpretable (arguably since it's the output of the critic model, it's not). In a re-write, I would maybe focus more on doing more direct comparisons between this new risk-based approach, vs. doing direct time-series modeling of the data (with no RL).
  * The authors situate their "explainability method" by discussing relevant literature on explainability methods like GradCAM that are commonly used in classification settings. Since they've formulated the problem as a time-series task, I'd actually argue that the most relevant comparable XAI methods are those that do interpretable time-series classification? Or that at minimum treat the task like it is time-series classification. See [1] for a list of examples.

[1] https://bmcmedinformdecismak.biomedcentral.com/articles/10.1186/s12911-020-1063-x

**Questions:**

Please see my questions listed under each of the weaknesses above.

---

### Note · Authors · 2024-11-25

**Comment:**

Thanks you to the reviewers for their detailed insightful comments.

We plan to take the reviews and work to address them more carefully for a later submission.

**Withdrawal Confirmation:**

I have read and agree with the venue's withdrawal policy on behalf of myself and my co-authors.